# Redo Pelvic Surgery and Combined Metastectomy for Locally Recurrent Rectal Cancer with Known Oligometastatic Disease: A Multicentre Review

**DOI:** 10.3390/cancers15184469

**Published:** 2023-09-08

**Authors:** Cian Keogh, Niall J. O’Sullivan, Hugo C. Temperley, Michael P. Flood, Pascallina Ting, Camille Walsh, Peadar Waters, Éanna J. Ryan, John B. Conneely, Aleksandra Edmundson, John O. Larkin, Jacob J. McCormick, Brian J. Mehigan, David Taylor, Satish Warrier, Paul H. McCormick, Mikael L. Soucisse, Craig A. Harris, Alexander G. Heriot, Michael E. Kelly

**Affiliations:** 1Department of Surgery, St. James’s Hospital, School of Medicine, Trinity College Dublin, D02 R590 Dublin, Ireland; 2Department of Surgery, Royal Brisbane and Women’s Hospital, Brisbane 4029, Australia; 3School of Medicine, University of Queensland, Brisbane 4072, Australia; 4Department of Surgery, Peter MacCallum Cancer Centre, Melbourne 3000, Australia; 5Department of Surgery, Hôpital Maisonneuve-Rosemont, Montreal, QC H1T 2M4, Canada; 6Department of Surgery, Mater Misericordiae University Hospital, D07 R2WY Dublin, Ireland; 7Trinity St. James Cancer Institute, D08 W9RT Dublin, Ireland

**Keywords:** locally recurrent rectal cancer, combined metastasectomy, recurrence

## Abstract

**Simple Summary:**

Due to advancements in surgical techniques, patients with recurrent rectal cancer and liver or lung secondaries are being increasingly considered for aggressive surgery. Our study reports outcomes for patients within this category undergoing simultaneous surgical removal of their cancer recurrence and secondaries across four centres worldwide. Our findings show that this option is feasible in a highly motivated group of patients who are well counselled on potential outcomes, including a high risk of cancer recurrence.

**Abstract:**

Introduction: Historically, surgical resection for patients with locally recurrent rectal cancer (LRRC) had been reserved for those without metastatic disease. ‘Selective’ patients with limited oligometastatic disease (OMD) (involving the liver and/or lung) are now increasingly being considered for resection, with favourable five-year survival rates. Methods: A retrospective analysis of consecutive patients undergoing multi-visceral pelvic resection of LRRC with their oligometastatic disease between 1 January 2015 and 31 August 2021 across four centres worldwide was performed. The data collected included disease characteristics, neoadjuvant therapy details, perioperative and oncological outcomes. Results: Fourteen participants with a mean age of 59 years were included. There was a female preponderance (n = 9). Nine patients had liver metastases, four had lung metastases and one had both lung and liver disease. The mean number of metastatic tumours was 1.5 +/− 0.85. R0 margins were obtained in 71.4% (n = 10) and 100% (n = 14) of pelvic exenteration and oligometastatic disease surgeries, respectively. Mean lymph node yield was 11.6 +/− 6.9 nodes, with positive nodes being found in 28.6% (n = 4) of cases. A single major morbidity was reported, with no perioperative deaths. At follow-up, the median disease-free survival and overall survival were 12.3 months (IQR 4.5–17.5 months) and 25.9 months (IQR 6.2–39.7 months), respectively. Conclusions: Performing radical multi-visceral surgery for LRRC and distant oligometastatic disease appears to be feasible in appropriately selected patients that underwent good perioperative counselling.

## 1. Introduction

Colorectal cancer is the third most commonly diagnosed neoplasm each year worldwide [1], with metastatic disease present in 15–20% of cases at time of diagnosis [2,3,4]. A further 25% will develop metastases during their treatment course [5,6]. The management of synchronous colorectal metastases has evolved significantly in recent times, due to advancements in surgery and perioperative support that allows the surgical treatment of previously ‘unresectable’ disease [7]. In patients with metastatic rectal cancer involving the liver or lung, long-term survival is possible when a multi-modality treatment strategy is employed in carefully selected patients [7,8,9,10,11,12].

Locally recurrent rectal cancer (LRRC) is defined as the recurrence, progression, or development of new sites of rectal tumour within the pelvis, occurring after previous surgical resection for rectal cancer [13]. The management of LRRC can be technically challenging, often requiring radical surgery [14,15]. This provides the best chance for long-term survival [16]. A 5-year survival rate is poor in these patients without surgery, as low as 4% with a median survival of six months [17,18,19]. The choice of management is dependent on multiple factors, such as prior therapy, the extent of the recurrence and the presence of distant disease [20].

Until recent years, radical multi-visceral resection in patients with LRRC had previously been reserved for those without metastatic disease outside the pelvis [21]. Less than half of those referred with LRRC typically proceed to an operation, with metastatic disease being the most common reason for avoiding major surgery [17]. However, ‘selective’ patients with limited oligometastatic disease (OMD) (involving the liver and/or lung) are increasingly being considered for radical resection. Favourable five-year survival rates have been demonstrated [7,8,9,11]. To date, these surgeries have tended to be staged resections of either the LRRC or oligometastatic disease site. Simultaneous multi-visceral resections have been reported in technically feasible cases (low-volume, favourable disease) and in patients with good performance status [12]. The potential benefits of performing a combined resection include a single anaesthetic and procedure, as well as an overall reduction in total length of hospital stay [12,22]. However, this approach increases the complexity of the surgical procedure and duration of surgery, raising concerns regarding perioperative safety and oncological outcomes of utilising this approach [23]. Ultimately, there are very few data in the literature on outcomes of patients undergoing simultaneous multi-visceral resection of LRRC with OMD [24,25,26,27].

Meticulous patient selection is essential when considering patients for this type of surgery, in order to both maximise positive outcomes and minimise morbidity in this cohort of patients [28]. Re-resection with combined metastasectomy should be reserved for medically fit patients with limited distant disease and pelvic symptoms retractable to conventional treatment [29]. Of course, it is imperative that patients are well counselled on potential survival outcomes and perioperative morbidity, in order to manage expectations for this highly selective treatment strategy [30].

The aim of our study is to provide vital data on both perioperative and oncological outcomes in patients undergoing synchronous or metachronous resection of LRRC with oligometastatic disease resection. These data will undoubtedly shed light on the feasibility and efficacy of this approach. In an era where personalised medicine and innovative treatment strategies are at the forefront of oncological research, combined LRRC resection and metastasectomy remains a pioneering endeavour.

## 2. Materials and Methods

### 2.1. Study Aim

This is a retrospective, multi-centre review investigating perioperative and oncological outcomes in patients undergoing resection for LRRC and OMD. In particular, we will report oligometastatic disease characteristics, histopathological margins, long-term outcomes and patterns of disease recurrence in this complex cohort of patients.

### 2.2. Statistical Analysis

Data were analysed using Statistical Package for the Social Sciences (SPSS) (Version 22, IBM, Chicago, IL, USA). Results were reported as mean +/− standard deviation (SD) as appropriate. Categorical variables were reported as proportions. Descriptive analysis was undertaken to report variable frequencies. Survival was estimated by the Kaplan–Meier method. The duration of survival for each case was defined as the time from the month of surgery to the month of death.

### 2.3. Study Design

Our study is a retrospective analysis of consecutive patients undergoing synchronous or metachronous resection of LRRC with oligometastatic disease between 1st January 2015 and 31st August 2021. Patients with histologically confirmed recurrent rectal cancer, following surgical treatment for index rectal cancer, undergoing exenterative surgery with concurrent metastectomy between 1 January 2015 and 31 August 2021 were included. Unless contraindicated, all included patients initially underwent neoadjuvant chemoradiotherapy prior to initial resection. Upon diagnosis, patients had ‘pseudo-neoadjuvant’ chemotherapy and were re-staged to exclude disease progression. Patients were excluded if their resection was carried out with palliative intent or if there were insufficient patient follow-up data available (minimum 30 days). Synchronous metastases were defined as those which were diagnosed prior to or within six months of initial diagnosis. Metachronous metastases were defined as those which presented more than six months after diagnosis. This comprehensive retrospective analysis spans a period of over six years, capturing a dynamic period during which the landscape of LRRC management underwent a notable transformation.

Data on patients who underwent surgical resection of LRRC with synchronous or metachronous resection of distant metastases meeting all of our inclusion/exclusion criteria were retrospectively collected from four international centres:Trinity St. James’s Cancer Institute, Dublin, Ireland (SJH).Peter MacCallum Cancer Centre, Melbourne, Australia (PMCC).Royal Brisbane and Women’s Hospital, Brisbane, Australia (RBWH).Hôpital Maisonneuve-Rosemont, Montreal, Canada (HMR).

For each of the included patients, the following data were reported. Patient demographics, including their age, sex, body mass index (BMI), American Society of Anaesthesiology (ASA) score, Charlson co-morbidity index (CCI). Disease characteristics included primary tumour location, oligometastatic disease location, metastasis size and number of metastases. Details of neoadjuvant therapy and ‘pseudo-neoadjuvant’ therapy, where available, were reported. Operative details were recorded and included type of exenterative surgery, type of oligometastatic disease resection, duration of surgery, estimated blood loss (EBL) as well as blood transfusion requirements. Histopathological details were recorded. These included margin status of both local recurrence and metastatic disease, lymph node yield and number of positive nodes. Post-operative details, such as length of stay, 30-day readmission rate, major complications (Clavien-Dindo > 2), re-intervention rate and overall mortality were recorded. Finally, long-term survival outcomes were reported (recurrence, overall survival).

### 2.4. Ethics

The treatments and research were in accordance with the ethical standards set by the Declaration of Helsinki and research was approved by independent ethical committees at each centre.

## 3. Results

### 3.1. Patient Demographics

Fourteen patients underwent synchronous or metachronous pelvic exenteration of LRRC with OMD. Synchronous metastases (n = 9) was defined as metastatic disease diagnosed before or within 6 months of initial resection. Metachronous metastases (n = 5) was defined as disease diagnosed at an interval more than 6 months after initial resection. Baseline participant characteristics are outlined in Table 1 and Table 2. Out of fourteen cases, 35.7% were male (n = 5) and the remaining 64.2% (n = 9) were female. Mean age at time of surgery was 59 +/− 12. The overall majority (92.8%, n = 13) underwent neoadjuvant chemoradiotherapy prior to primary tumour resection. Upon diagnosis of recurrence, all patients had ‘pseudo-neoadjuvant’ chemotherapy over 3 months. They were subsequently restaged to ensure no disease progression prior to proceeding with their multi-visceral resection. No patients were re-irradiated.

### 3.2. Disease Characteristics

Disease characteristics are outlined in Table 3. The liver accounted for 64.3% (n = 9) of oligometastatic disease. 28.5% (n = 4) were in the lung and 7.1% (n = 1) involved both the lung and liver. Mean liver and lung tumour sizes were 2.9 cm +/− 1.5 cm and 1.2 cm +/− 0.51 cm, respectively. The mean number of metastatic tumours present in each patient was 1.5 +/− 0.5.

### 3.3. Perioperative Details

Operative details are outlined in Table 4. Overall, 73.3% (n = 11) of patients underwent a modified pelvic exenterative surgery, with the remainder (n = 3) undergoing total exenteration. Of the fifteen oligometastatic disease resections, 66.6% (n = 10) were of liver lesions and the remaining 33.3% (n = 5) were of lung metastases. Oligometastatic disease resections of the liver included wedge resection (n = 2), segmentectomy (n = 2) and partial hepatectomy (n = 6). Oligometastatic disease resections of the lung included wedge resection (n = 3) and lobectomy (n = 2).

Mean operating time was 431 +/− 85 min with overall mean estimated blood loss in being 1287 +/− 792 mL. Only two patients required blood transfusions, one requiring 2 units packed red cells and the other requiring 4 units. Median length of hospital stay (LOS) was 14 days.

Overall mean operating time for LRRC resections was 536 +/− 197 min. Overall mean EBL for LRRC resections was 1400 +/− 544 mL. Eight patients required a blood transfusion, mean transfusion requirement of 2.15 +/− 2.03 units. Average LOS in these patients was 18 +/− 10 days.

There were no 30-day perioperative mortality to report. The only major morbidity was a perineal wound complication requiring return to theatre for a debridement.

### 3.4. Histopathology

Histopathological outcomes are demonstrated in Table 5. R1 margins on the LRRC exenteration specimen occurred in four cases (28.6%). There were no positive margins in the metastectomy specimens. Mean lymph node yield was 11 +/− 6.9.

### 3.5. Long-Term Outcomes

Long-term outcomes are outlined below in Table 6. Overall recurrence was 70% (n = 7) out of ten cases reporting data on that outcome. At follow-up, the median disease-free survival and overall survival were 11 (IQR 4–17) and 20 (IQR 6–39) months, respectively. Out of the seven patients with recurrence, four and three patients had R1 and R0 exenterative resections, respectively. All included patients had R0 metastatic margins.

## 4. Discussion

Despite up to 50% of patients with LRRC having metastatic disease at diagnosis of recurrence, there are scarce data in the literature evaluating outcomes of these patients undergoing synchronous or metachronous resection of their recurrence and their metastases [14,15]. In this study we investigated perioperative, histopathological and oncological outcomes of this cohort of patients, aiming to determine the feasibility of this approach in the treatment of LRRC with oligometastatic disease. Our study demonstrated reasonable overall and disease-free survival rates in this cohort of patients. Of course, redo pelvic surgery with combined metastasectomy is not suitable for all patients, and should be reserved for a highly selective, motivated group of patients.

The use of radical resection in the treatment of LRRC has been well established [16,31]. Previous studies have demonstrated a 5-year overall survival of 4% and median survival of 6–7 months in those treated without radical surgery [17,18,19]. Tanaka et al. performed a retrospective review of 70 consecutive patients in their centre who underwent surgical resection of LRRC with curative intent, ten of which had synchronous distant metastases (SDM) [32]. Of the ten patients with SDM, four had simultaneous resection and the remaining six had a staged procedure. The authors reported an R0 rate of 80% (n = 8) and an overall recurrence rate of 90% (n = 9) (four of local and six distant recurrences). 5-year overall survival and 3-year recurrence-free survival rates were 40.5% and 10%, respectively. These findings were not dissimilar to our own study, for which we observed R0 rates and overall recurrence rates of 70%. The study reported an overall survival of 47.4 +/− 29.8 months, significantly higher than our findings (25 +/− 19.8 months). We observed a similar disease-free survival in our cohort compared to that described by Tanaka et al. (12.3 +/− 9.9 months vs. 12.3 +/− 6.92 months). Both studies demonstrated acceptable oncological and perioperative outcomes in the treatment of LRRC with SDM.

Controversy surrounds both the choice of patient, and choice of timing in synchronous metastatic disease resection in the treatment of LRRC [33,34]. Molecular profiling is increasingly being used to identify CRC patient subgroups who are resistant to conventional systemic treatment. A recent meta-analysis [34] has shown molecular profiling can help to identify the patients at risk of early failure after metastasectomy. Patient selection is essential to optimal outcomes, and so molecular profiling in these cases should be used where available. RAS and BRAF mutations were associated with a negative prognosis, whereas TP53 and PIK3CA mutations had little effect on long-term outcomes. The biological status of each tumour may provide the basis for personalised treatment, and mutational status should be considered when selecting both timing and approach of surgical resection.

Slow-growing recurrent rectal tumours may cause significant pain refractory to conventional analgesia, secondary to carcinomatous invasion of the lumbosacral plexus or sciatic, obturator, pudendal or spinal nerves [35,36]. Surgical resection may offer these patients an improved quality of life in addition to an improved disease-free and overall survival time period [29]. Of course, surgical resection imposes its own morbidity on patients and the decision to proceed with radical surgery should not be taken lightly. Meticulous patient selection is imperative, when considering patients for this type of surgery, in order to maximise positive outcomes while avoiding harmful or futile interventions [28]. Medically fit patients with limited distant disease and pelvic symptoms retractable to conventional analgesia could be offered re-resection with metastasectomy, following MDT discussion, provided that they were well counselled in potential survival outcomes [29].

Regarding the timing of metastasectomy, while simultaneous resection has the advantage of a single anaesthetic, operation and reduced total length of stay, resecting distant metastases at least two months prior to exenterative surgery may avert the need for radical resection in cases where the metastatic disease is deemed uncurable [32]. The optimal perioperative treatment timing also remains unresolved, with more centres now opting for a neoadjuvant approach over standard adjuvant chemotherapy due to poor compliance rates [33]. Further research is required to determine the optimal approach, taking into consideration not only oncological outcomes, but also patient quality of life [30].

Our study, while providing valuable insights into the management of locally recurrent rectal cancer (LRRC) with synchronous distant metastases, is not without its limitations. The most significant constraint is the relatively small sample size, despite recruiting from multiple international centres. The scarcity of such cases limits the number of eligible patients available for study inclusion. Consequently, our study lacked the statistical power needed to confidently compare outcomes between patients undergoing synchronous and metachronous resection of distant metastases. A larger sample size would have allowed for more robust statistical analyses, potentially revealing subtle differences in outcomes and providing clearer guidance for clinicians facing these challenging treatment decisions. Furthermore, the retrospective nature of our data collection introduced additional limitations. In some cases, the data necessary to assess specific intended outcomes were missing or incomplete, resulting in further reduction in our case numbers for certain analyses. This limitation underscores the importance of prospective, multi-centre collaborative research.

Despite these limitations, our study adds valuable data to the already scarce literature on patients with LRRC undergoing synchronous or metachronous resection of metastases. LRRC with synchronous distant metastases remains a challenging clinical scenario, and our study provides valuable real-world outcome data and insights that can guide clinicians in their decision-making processes.

## 5. Conclusions

Performing radical surgery for LRRC and distant oligometastatic disease resection appears to be a feasible option in a highly selective and motivated cohort with good perioperative counselling. Our study contributes essential insights into the complex management of patients with LRRC undergoing synchronous or metachronous resection of their local recurrence and distant oligometastatic disease. Operative outcomes appeared favourable when a combined resection was performed. Further research is required with larger patient numbers to determine whether simultaneous or staged resection of metastases offers the best perioperative and oncological outcomes for this difficult-to-treat group of patients.

In an era where personalised medicine and innovative treatment strategies are at the forefront of oncological research and patient care, our study reaffirms the necessity of meticulous patient selection and adequate patient counselling. Quality of life should be prioritised and patient expectations should be managed accordingly when considering patients for this radical treatment approach.

## Figures and Tables

**Table 1 cancers-15-04469-t001:** Baseline participant characteristics.

Patient	Age (Years)	Gender	BMI	ASA	CCI
1	48	Female	32	2	6
2	58	Female	35	2	7
3	68	Female	22	2	8
4	40	Male	40	2	6
5	62	Female	NA	2	8
6	55	Male	27	2	7
7	41	Female	22	1	6
8	70	Female	25	3	9
9	71	Female	27	2	7
10	65	Female	25	2	6
11	64	Male	27	3	7
12	73	Male	26	3	7
13	76	Female	22	3	8
14	46	Male	32	2	6

(BMI: Body mass index, ASA: American Society of Anaesthesiology score, CCI: Charlson co-morbidity index).

**Table 2 cancers-15-04469-t002:** Baseline participant characteristics (mean +/− SD).

Age (mean +/− SD)	59 +/− 12
Gender (Male:Female)	5:9
BMI (mean +/− SD)	27.8 +/− 5.4
ASA(mean +/− SD)	2.2 +/− 0.6
Charlson co-morbidity index (mean +/− SD)	7 +/− 0.96
Neoadjuvant treatment n (%)	13 (92.8%)

**Table 3 cancers-15-04469-t003:** LRRC and OMD characteristics.

Oligometastatic Disease Location n = 14	
Liver	9
Lung	4
Both	1
**Oligometastatic disease size (cm)**	
Liver (mean +/− SD)	4.03 +/− 1.82
Lung (mean +/− SD)	1.7 +/− 0.96
No. metastatic tumours (mean +/− SD)	1.5 +/− 0.85

**Table 4 cancers-15-04469-t004:** Operative details.

LRRC Operation n = 14	
Pelvic exenteration	14
Total	3
Modified	11
**Oligometastatic disease resection** **n = 15**	
Liver	10
Wedge resection	2
Segmentectomy	2
Partial hepatectomy	6
Lung	5
Wedge resection	3
Lobectomy	2

**Table 5 cancers-15-04469-t005:** Histopathological outcomes for both procedures.

Surgical Margins	
LRRC surgery	n = 14
R0 n (%)	10 (71.4%)
R1 n (%)	4 (28.6%)
Oligometastatic disease surgery	n = 14
R0 n (%)	14 (100%)
R1 n (%)	0
Lymph node yield (mean +/− SD)	11 +/− 6.9
Positive nodes n (%)	4 (28.6%)

**Table 6 cancers-15-04469-t006:** Long-term outcomes and patterns of disease recurrence.

Oncological Outcomes n = 10	
Recurrence n (%)	7 (70%)
Disease-free survival (months) (mean +/− SD)	12.3 +/− 9.9
**n = 14**	
Overall survival (months) (mean +/− SD)	25.9 +/− 19.8
**Recurrence Location** **n = 7**	
Lung	2
Liver	2
Lung and Liver	1
Sacrum	1
Retroperitoneal lymph nodes	1

## Data Availability

Data are contained within the article.

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
