# Peer review of "Redo Pelvic Surgery and Combined Metastectomy for Locally Recurrent Rectal Cancer with Known Oligometastatic Disease: A Multicentre Review"

_cancers, 2023, doi:10.3390/cancers15184469_

Round 1

Reviewer 1 Report

This is a retrospective, multi-institutional study evaluating the short and long-term outcomes in patients affected by LRRC with oligometastatic disease. Despite the topic is timely and relevant, several major criticisms needs to be addressed:

1- The manuscript, especially in the Methods section, is not properly written. The study design and methodology cannot be just a list of bullet points, but should be carefully and clearly described.

2- The Authors state that synchronous vs. metachronous oligometastatic disease resection will be compared, but such analysis never appears in the Results. Furthermore, it is no stated at all how many patients were treated with synchronous resection vs staged resection. Moreover, there is also a lacking in details about neoadjuvant treatment: how many re-irradiations? How many patients treated by induction chemotherapy? Which regimens were used?

3- No surgical details are provided for LRRC resection (just named “exenterative surgery”) and above all for metastatic disease surgery: how many hepatectomies? How many wedge resections? How many segmentectomies? Etc.

4- The most relevant criticism, is the proposed take-home message itself, which is absolutely not supported by findings. Indeed, the Authors concluded that surgical treatment of LRRC patients with oligometastatic disease is feasible and further assessed in larger studies. However, from the study a 70%  re-recurrence rate emerged, with a  poor median disease-free survival. Interestingly, the R0 rate for metastases was 100%, but the re-recurrences were mainly distant disease relapses. These data suggest that in this setting currently there is a limited role for surgery, as the disease is systemic and a proper resection of both LRRC and metastases still does not provide a convincing modality of cure of these patients, which is not in accordance with the conclusions stated by the Authors.

5- Moreover it’s not clear how many out the seven recurring patients underwent a R0 or a R1 exenterative surgery . These details could have a paramount relevance in order to select patients candidated to  such a complex and demolitive resections and should be properly discussed within the manuscript.

Reviewer 2 Report

The article is on a very interesting topic, well structured, I enjoyed reading it.

The tables should be revised according to the Journal's requirements, especially Table 4 (not clear).

Reviewer 3 Report

First of all, thank you so much for involving me in reviewing this manuscript.

Very interesting and topical topic always of great study and debate. Above all today with the development of new techniques that allow to broaden the indications and push into the surgical treatment of cases that previously were only of oncological relevance.

Clear and easily understood English language with easy reading.

Adequate and recent bibliography with a number of valid references.

Clear and understandable tables and images.

For me there are no changes to be made and the manuscript can be published.

Round 2

Reviewer 1 Report

the paper has been properly revisited.

Many of the hot points underlined have been clarified. 

presently it can be considered for publication in accordance with the programs  of the journal.